

# An explicit solution for calculating optimum spawning stock size from Ricker's stock recruitment model

Mark D. Scheuerell

Fish Ecology Division, Northwest Fisheries Science Center, National Marine Fisheries Service, National Oceanic and Atmospheric Administration, Seattle, WA, United States

## ABSTRACT

Stock-recruitment models have been used for decades in fisheries management as a means of formalizing the expected number of offspring that recruit to a fishery based on the number of parents. In particular, Ricker's stock recruitment model is widely used due to its flexibility and ease with which the parameters can be estimated. After model fitting, the spawning stock size that produces the maximum sustainable yield ($S_{MSY}$) to a fishery, and the harvest corresponding to it ($U_{MSY}$), are two of the most common biological reference points of interest to fisheries managers. However, to date there has been no explicit solution for either reference point because of the transcendental nature of the equation needed to solve for them. Therefore, numerical or statistical approximations have been used for more than 30 years. Here I provide explicit formulae for calculating both $S_{MSY}$ and $U_{MSY}$ in terms of the productivity and density-dependent parameters of Ricker's model.

Subjects Ecology, Mathematical Biology
Keywords Ricker model, Maximum sustainable yield, MSY, Stock-recruit, Spawner, Harvest

## INTRODUCTION

One of the most difficult problems in the assessment of fish stocks is establishing the relationship between the spawning stock and subsequent recruitment (*Hilborn & Walters, 1992*). Stock-recruitment models have been used for decades in fisheries management as a means of formalizing this relationship (*Beverton & Holt, 1957*; *Ricker, 1954*). Over time, a variety of functional forms have emerged to capture varying assumptions about depensatory and compensatory mortality (*Hilborn & Walters, 1992*). In a classroom setting, deterministic versions of the models provide useful constructs for teaching about management reference points such as maximum sustained yield (MSY).

In particular, Ricker's stock recruitment model (*Ricker, 1954*; *Ricker, 1975*) is one of the most widely used models to describe the population dynamics of fishes, such that

$$R = \alpha S e^{-bS}, \tag{1}$$

$R$ is the number of recruits produced, $S$ is the number of spawners, $\alpha$ is the dimensionless number of recruits per spawner produced at very low spawner density, and $b$ is the strength of density dependence (units: spawner$^{-1}$). It is common to substitute $\alpha = e^a$ into

Corresponding author
Mark D. Scheuerell,
mark.scheuerell@noaa.gov

Eq. (1) and rewrite it as

$$R = Se^{a-bS}. \tag{2}$$

To make the model reflect a stochastic process, Eq. (2) is typically multiplied by a log-normal error term, so that

$$R = Se^{a-bS}e^{\varepsilon}, \tag{3}$$

and $\varepsilon$ is a normally distributed error term with a mean of $-1/2\,\sigma$ and variance $\sigma$. This non-zero mean ensures that $a$ is interpreted as the mean recruits per spawner rather than the median (Hilborn, 1985). Part of the model's popularity is due to the relative ease with which its parameters are estimated. After log transformation, Eq. (3) is typically rewritten as

$$\ln(R/S) = a - bS + \varepsilon, \tag{4}$$

and the parameters are estimated via simple linear regression. I note here that estimation of the parameters via a simple observation-error model like (4) can lead to substantial biases in $a$ and $b$ if the sample size is low ($n \leq 10$) due to autocorrelation in the residuals $\varepsilon$ (Walters, 1985).

Once the model has been fit to data and any necessary bias corrections made, the parameters can be used to derive various biological reference points of interest to fisheries managers. Some of these metrics are rather trivial to compute. For example, the spawning stock size leading to maximum recruit production ($S_{\mathrm{MSR}}$) is simply $1/b$. However, other reference points are much less straightforward to calculate. In particular, the spawning stock expected to produce the maximum sustainable yield ($S_{\mathrm{MSY}}$) under deterministic dynamics is of common interest.

To find $S_{\mathrm{MSY}}$, I express the yield ($Y$) as

$$Y = R - S = Se^{a-bS} - S, \tag{5}$$

and then take the derivative of $Y$ with respect to $S$:

$$\frac{dY}{dS} = (1 - bS)e^{a-bS} - 1. \tag{6}$$

$S_{\mathrm{MSY}}$ is then determined by setting Eq. (6) to zero and solving for $S$. Upon initial inspection, however, there does not appear to be an explicit solution to this equation in terms of $S$, and therefore $S_{\mathrm{MSY}}$ is typically solved "by trial" (Ricker, 1975) with some form of gradient method (e.g., Newton's as in Hilborn, 1985).

To simplify this issue for common applications, Hilborn (1985) developed a simple model whereby the ratio of spawning stock size at MSY to that at the unfished equilibrium ($S_{\mathrm{MSY}}/S_r$) is a linear function of the parameter $a$. Specifically, for $0 < a \leq 3$ he estimated that

$$\frac{S_{\mathrm{MSY}}}{S_r} = \frac{S_{\mathrm{MSY}}}{(a/b)} = 0.5 - 0.07a, \text{ and} \tag{7a}$$

$$S_{\mathrm{MSY}} = \frac{a(0.5 - 0.07a)}{b}. \tag{7b}$$

Although this approximation is very useful due to its simplicity, there is no underlying fundamental support for the statistical form of the relationship.

## METHODS

Here I make use of the Lambert W function, $W(z)$, to demonstrate an explicit solution to Eq. (4) that precludes the need to estimate $S_{\mathrm{MSY}}$ via numerical methods or *Hilborn*'s (*1985*) linear approximation. This function has been used for explicit solutions to Roger's random predator equation in ecology (*McCoy & Bolker, 2008*) and susceptible-infected-removed (SIR) models in epidemiology (*Reluga, 2004*; *Wang, 2010*). Specifically, $W(z)$ is defined as the function that satisfies

$$W(z)e^{W(z)} = z \tag{8}$$

for any complex number $z$ (Lambert 1758 and Euler 1783 as cited in *Corless et al., 1996*). Here we are interested only in real values, however, so I replace $z$ with $x$ and note that $W(x)$ is only defined for $x \geq -1/e$ (*Corless et al., 1996*). Furthermore, this function is not injective and has two values for $-1/e \leq x \leq 0$, but as I show below, we are concerned only with the region where $x > 0$ and $W(x)$ is a singular, non-negative value.

I begin my explicit solution of $S_{\mathrm{MSY}}$ by setting Eq. (6) to zero, such that

$$(1 - bS_{\mathrm{MSY}})e^{a - bS_{\mathrm{MSY}}} = 1. \tag{9}$$

After rearranging terms and multiplying both sides by $e$, we arrive at

$$(1 - bS_{\mathrm{MSY}})e^{1 - bS_{\mathrm{MSY}}} = e^{1-a}. \tag{10}$$

At this point I note the relationship between Eqs. (10) and (8), with $1 - bS_{\mathrm{MSY}} = W(z)$ and $e^{1-a} = z$. Therefore, we can write

$$1 - bS_{\mathrm{MSY}} = W\left(e^{1-a}\right), \text{ and hence} \tag{11}$$

$$S_{\mathrm{MSY}} = \frac{1 - W\left(e^{1-a}\right)}{b}. \tag{12}$$

We now have an explicit solution for $S_{\mathrm{MSY}}$ that depends only on the parameters $a$ and $b$ from Eq. (2). As mentioned above, $W(x)$ is only defined for $x \geq -1/e$, which does not pose any problems here because $x = e^{1-a} > 0 \ \forall \ a \in \mathbb{R}$. For visualization purposes, I show a plot of $W(e^{1-a})$ versus $a$ in Fig. 1.

We can also derive an explicit formula for calculating the fraction of the return harvested at $S_{\mathrm{MSY}}$, which I call $U_{\mathrm{MSY}}$. As *Ricker (1975)* shows,

$$U_{\mathrm{MSY}} = bS_{\mathrm{MSY}}, \tag{13}$$

and therefore substituting (12) into (13) gives

$$U_{\mathrm{MSY}} = 1 - W\left(e^{1-a}\right). \tag{14}$$

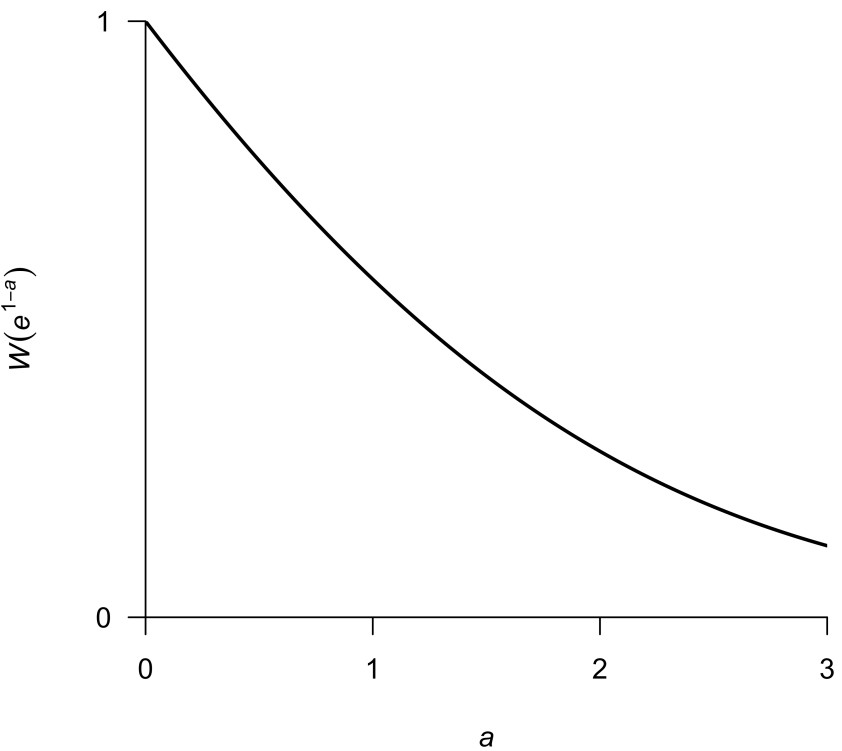

**Figure 1** Plot of $W(e^{1-a})$ over a range in values of $a$ typically encountered in fisheries.

In practice $W(x)$ may be approximated numerically using some form of gradient method. *Corless et al. (1996)* recommend Halley's method, with the update equation given by

$$w_{j+1} = w_j - \frac{w_j e^{w_j} - x}{e^{w_j}(w_j+1) - \frac{(w_j+2)(w_j e^{w_j} - x)}{2w_j+2}}. \tag{15}$$

I used an initial guess of $w_0 = 3/4 \ln(x+1)$ based on the shape of $W(x)$ over the range of $a$ typically considered in fisheries research (i.e., $0 < a < 3$ as in *Hilborn, 1985*). If, however, one must estimate $W(x)$ numerically, then one should ask whether doing so is, in fact, computationally faster. Therefore, as a test I randomly selected 1,000 values each for $0 < a \le 3$ and $10^{-5} \le b \le 10^{-3}$, and then solved for $S_{MSY}$ using both Newton's method as suggested by *Ricker (1975)*, and Halley's method as in Eq. (15).

## RESULTS AND DISCUSSION

Recent analyses have relied on estimating $S_{MSY}$ via *Hilborn*'s (*1985*) linear approximation when calculating optimal yield profiles (*Fleischman et al., 2013*) or the effects of observation error on biases in parameter estimates (*Su & Peterman, 2012*). On the other hand, solving for $S_{MSY}$ using $W(x)$ and Halley's method is not only convenient; it also offers an appreciable computational advantage over the standard Newton method. Although both methods converged in less than 10 iterations during my test, Halley's method was always faster and less variable overall (Fig. 2). Therefore, estimating $S_{MSY}$ via Halley's method

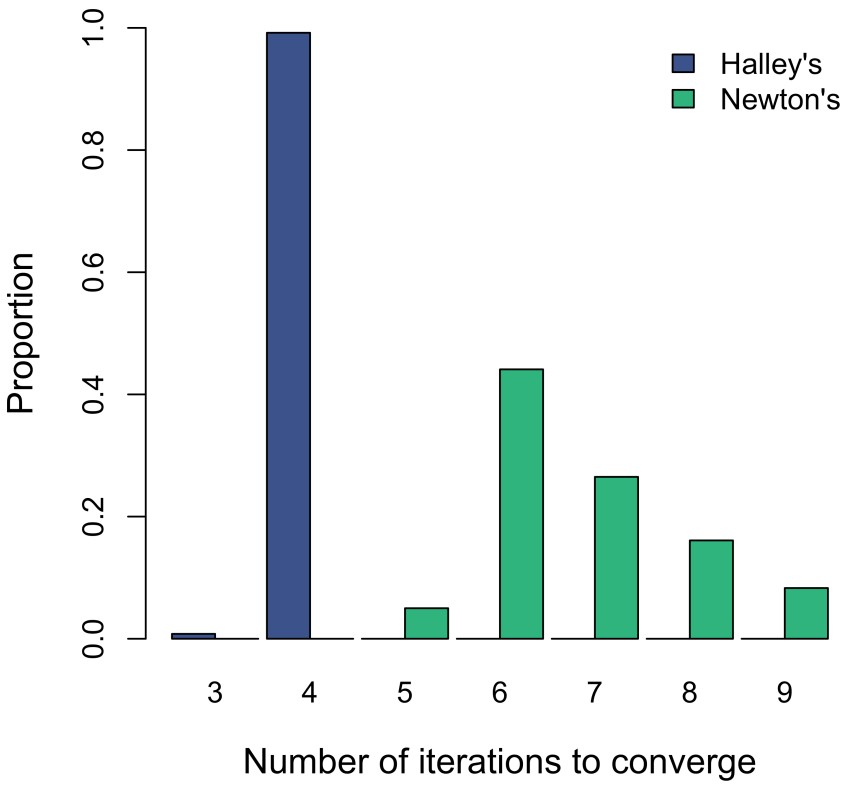

**Figure 2** Histograms showing the distribution of the number of iterations that each of the two numerical methods takes to converge to $S_{MSY}$ using a threshold of $10^{-6}$.

might save significant time in applications such as management strategy evaluations that are much more computationally intensive than a simple one-case solution.

Although implementing Eq. (15) may seem a bit daunting to individuals less familiar with numerical methods, a variety of contemporary software packages (e.g., MATLAB, R) include built-in functions to calculate $W(x)$ directly. This means that anyone using a personal computer to estimate the parameters in a Ricker model can easily estimate $S_{MSY}$ from Eq. (12) as I demonstrate in Table 1; I show the results from my R implementation for a range of $a$ and $b$ in Fig. 3. For those preferring to use Microsoft Excel, there is no built-in function to calculate $W(x)$, but I have implemented Eq. (15) as the VBA function 'LAMBERTW' in the Microsoft Excel add-in file 'LambertWfunc.xlam' (see Fig. S1 for download and install instructions).

Here I have outlined a new method to easily calculate $S_{MSY}$ from the productivity ($a$) and density-dependent ($b$) parameters in a Ricker model using readily available functions in several software packages. This method is much more straightforward than trying to solve for $S_{MSY}$ using numerical methods and should be useful in many classroom settings. Although there could be some utility in actually going through the exercise of numerically deriving the answer, it is rare nowadays, for example, for anyone to code a random number generator because of their ubiquitous implementation in standard software. In addition, the explicit analytical solution is closed-form with respect to the special functions, and

**Table 1** **Example code for directly calculating $S_{MSY}$ in R, Matlab, and Excel; the values for $a$ and $b$ were chosen arbitrarily.** Note that the R code requires the 'gsl' package to be installed, and the Excel code requires the 'LAMBERTW' function contained in the Excel Add-in file LambertWfunc.xlam.

| Software | Code example | | |
|---|---|---|---|
| R | ```> library(''gsl'')```<br>```> a = 1```<br>```> b = 5e−4```<br>```> Smsy = (1 − lambert_W0(exp(1−a))) / b``` | | |
| MATLAB | ```>> a = 1```<br>```>> b = 5e−4```<br>```>> Smsy = (1 − lambertw(exp(1−a))) / b``` | | |
| Excel | | A | B |
| | 1 | **a** | 1 |
| | 2 | **b** | 5e−4 |
| | 3 | **Smsy** | = (1 − LAMBERTW(EXP(1 − B1))) / B2 |

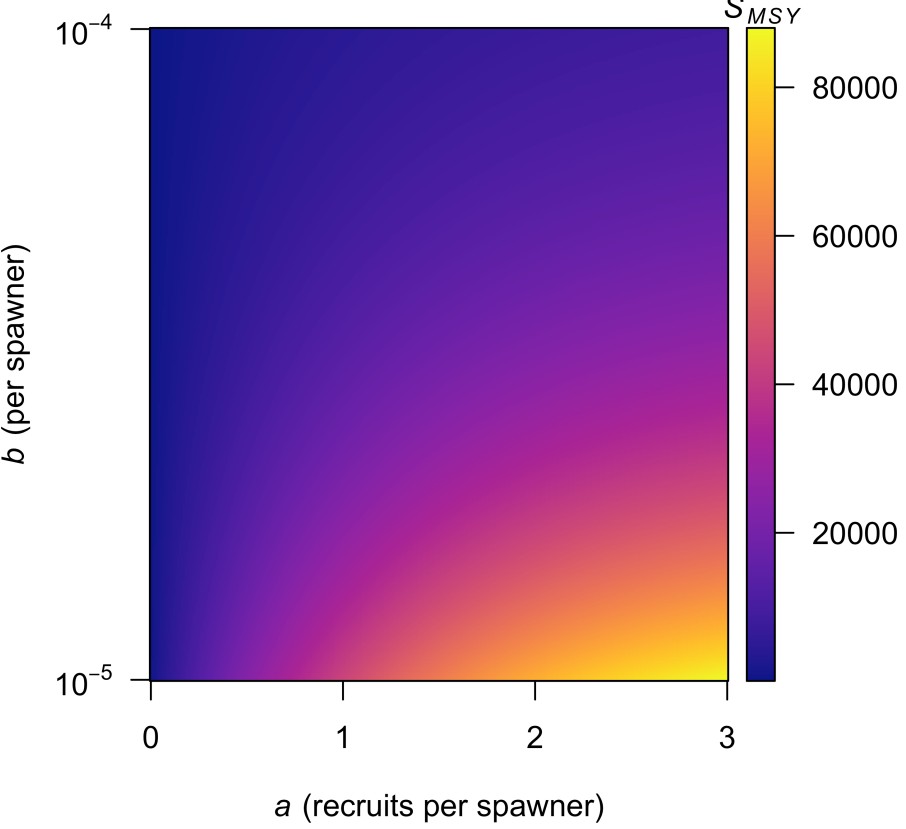

**Figure 3** **Contour plot showing values of $S_{MSY}$ for combinations of the $a$ and $b$ parameters in Eq. (2).**

therefore precludes the need to estimate $S_{MSY}$ via *Hilborn*'s (*1985*) approximation. Thus, due to the speed and ease with which these new equations are calculated, I recommend that practitioners use them for the estimation of $S_{MSY}$ and $U_{MSY}$ in lieu of those listed in Appendix III of *Ricker (1975)* and Table 7.2 of *Hilborn & Walters (1992)*.

## ACKNOWLEDGEMENTS

I thank Jim Thorson, Jason Link, Brian Kennedy, Olaf Jensen, Ray Hilborn, Tim Essington, Curry Cunningham, and Trevor Branch for helpful discussions and comments on the manuscript.

### Funding

The author received no external funding for this work.

### Competing Interests

The author declares there are no competing interests.

### Author Contributions

- Mark D. Scheuerell conceived and designed the experiments, analyzed the data, contributed reagents/materials/analysis tools, wrote the paper, prepared figures and/or tables, reviewed drafts of the paper.

### Data Availability

University of Washington faculty server: http://faculty.washington.edu/scheuerl/LambertWfunc.xlam.

### Supplemental Information

Supplemental information for this article can be found online at http://dx.doi.org/10.7717/peerj.1623#supplemental-information.

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
