# Peer review of "An explicit solution for calculating optimum spawning stock size from Ricker’s stock recruitment model"

_PeerJ, doi:10.7717/peerj.1623_

## Round 0.1 · original submission · Minor Revisions

The reviews are straight-forward and the suggested changes will improve the paper. I got another comment but not a review from a notable stock-recruit expert who said that he had seen an earlier version of your paper and that the math presented in your paper was done "years ago" by Ray Hilborn. I could not verify that and I think your math is an improvement in my experience with s/r calculations. But, please take another look at your citations of Ray's work and try to deal with the criticism. At least give me your take on it in your rebuttal.

·

Basic reporting

This submission is well written, succinct, clear, and will be easily understood by all readers, even those with limited quantitative background. Necessary references are made to previous work on stock-recruitment modeling, and Smsy/MSY approximation methods. I found the figures relevant to the text, and the exploration of the necessary number of iterations to convergence for approximations using both Newton-Raphson and Halley's methods quite insightful and helpful when determining which to implement in other software platforms without Lambert functions (i.e. JAGS).

Experimental design

Research described within this submission is both novel and quite useful from a practical perspective. The author clearly defines the research goal, identifies why the current method is superior to linear approximations or alternative iterative methods, and provides a logical series of equations to justify the validity of the Lambert approximation method. All equations, transformations, and derivations are correct as written, and can easily be reproduced.

Validity of the findings

The key finding of this submission, namely the efficacy of a new closed-form analytical solution for calculating Smsy from Ricker parameters using the Lambert function, is clearly demonstrated.

Additional comments

Several specific comments regarding wording and sentence structure. These are completely subject and should be accepted/rejected at the author's discretion:
Abstract
Line 3: “between parental ABUNDANCE and offspring PRODUCTION”
Line 12: Last line change “density-dependent parameters FROM Ricker’s…” to “density-dependent parameters OF Ricker’s…”
Introduction
Line 44-45, restructure to have qualifier at beginning “However, other reference points…”
Methods
Line 66: Change “Hilborn’s (1985) statistical model”, to “Hilborn’s (1985) linear approximation”
Line 93: might be worthwhile to indicate that x=z, by simply changing to: “In practice W(x), where x=z=exp(1-a), may be approximated…”

·

Basic reporting

The paper represents a novel way to solve for fundamental harvest parameters for fishery management models that have been used for more than half of a century. The manuscript is relatively straightforward and extremely well written and of applied interest. Furthermore it represents a comprehensive yet finite topic of interest that will be of interest to both managers as well as professors interested in teaching the quantitative tools in fisheries biology.

I think a weakness of the manuscript as it sits is that the author does not use an example from the literature to demonstrate the utility of the his approach nor in the discussion discuss the comparative weight of being able to quantify Smsy vs. having the dataset that would allow you to confidently use or apply the Ricker model in any meaningful way.

Experimental design

The mathematics that the authors detail are robust and are covered at a level that allows a general reader to follow along with little trouble. The author describes the model construction thoughtfully and precisely.

Figure 3 is not well set up or utilized or presented. The author does not establish the relevance of convergence time or provide any reference for the single digit iterations and thresholds required to converge, other than saying that "methods were remarkably quick". Perhaps a better visual presentation of the convergence process. Finally, the Halley's method is confusing in its presentation, am I to assume that for 1000 iterations of and b, convergence occurred at 4 iterations all but one time (the outlier)? There is no variation? Is there significance to this that can be discussed?

Validity of the findings

It would appear that the findings conform to the expectations of the journal. They are clearly stated and represent a unified publishable unit. The results are clear and well written. The findings will be useful in terms of its novel application of the Lambert W function, and again, as a genuinely useful tool for understanding the function of the Ricker model in advanced fisheries classes.

Additional comments

A few suggestions to improve some aspects of manuscript follow.

I think that the major improvement would be, as discussed above, an example of the application of the approach. Furthermore, the author ignores all discussion of the difficulties and issues of actually applying and using Smsy. Although a full treatment of this is beyond the scope of this research paper, the findings seem unbalanced without at least some minimal coverage of the challenges of applying Smsy (and therefore Umsy) in real world situations. As it stands, the results and discussion section appear light and repetitive to earlier text and would benefit from a fuller discussion of context and application.

Author should discuss when and why values for Smsy from Figure 2 are different from solving by numerical methods.

Line 2 and 3 - Rewrite second sentence so that it adds additional context otherwise it seems as an unnecessary restating of the first sentence.

Line 33 - is the author referring to the error term when saying the "non-zero mean". This is not clear to me.

Line 45 - awkward to end sentence in "however".

Line 57 - awkward wording of "relating the ratio...to...to..." Could be made clearer.

Line 97 - Is there and error in the wording, "Although implementing equation..."?

Line 126 - "...practitioners use them for [the estimation of] Smsy...."

Figure 2 - If you are going to assign units to b in line 27 (per spawner), then this should be included on the y axis. And my preference would be to list the numbers out, (0.0004) instead of the scaling function on the axis label.

---

## Round 0.2 · accepted · Accept

I really appreciated your thorough rebuttal. Good job on the revision.